# OpenReview forum: "Benchmarking Correctness and Security in Multi-Turn Code Generation"
_ICLR.cc/2026/Conference — Submitted to ICLR 2026_

### Official Review · Reviewer_2jB7 · 2025-10-26

**Soundness:** 2
**Presentation:** 3
**Contribution:** 2
**Rating:** 4
**Confidence:** 4

**Summary:**

The paper introduces MT-Sec, the first benchmark that systematically evaluates both functional correctness and security in multi-turn coding scenarios. It employs a synthetic data pipeline that transforms existing single-turn tasks into semantically aligned multi-turn interaction sequences, and comprehensively evaluates 32 open- and closed-source models as well as three agent-scaffolding frameworks on MT-Sec.
In addition, this work is the first to assess models’ ability to generate correct and secure code diffs, providing new insights into incremental code-generation safety.

**Strengths:**

1. The benchmark’s three synthetic formats—Expansion, Editing, and Refactoring—are practical and realistic, capturing common multi-turn coding workflows.
2. The inclusion of a manual verification step to ensure the validity of multi-turn requests makes the test case generation process reliable and credible.
2. The benchmark supports multiple programming languages and covers a wide range of CWEs, enhancing its comprehensiveness and generality.
3. The experiments are extensive, evaluating a large number of LLMs, which makes the empirical analysis thorough and convincing.

**Weaknesses:**

The comprehensive and large-scale experiments are appreciated. However, since the overall pipeline design is relatively straightforward and both the seed code and test suites are derived from existing benchmarks (e.g., SecCodePLT and BaxBench), the technical and methodological novelty appears somewhat limited.

Given that the core contribution lies in transforming single-turn tasks into multi-turn code generation, its scope appears limited to only three interaction types—Expansion, Editing, and Refactoring. Exploring a broader range of interaction patterns could further strengthen the benchmark’s coverage and realism.

**Questions:**

1. Do the authors plan to extend MT-Sec beyond the three existing formats (Expansion, Editing, Refactoring) to include more realistic multi-turn coding scenarios, such as debugging cycles, version recovery/recall, or other scenarios?

2. Why do some results in Table 1 show large discrepancies among the three categories (MT-Expansion, MT-Editing, and MT-Refactor)? Could the authors explain the distinct challenges or characteristics that lead to such performance gaps across these task types?
Additionally, would it be appropriate to incorporate a difficulty hierarchy across tasks to better reflect the varying complexity levels of different multi-turn interactions?

---

> ### Author Response · Authors · 2025-11-21
> **Author Rebuttal (Part-1)**
>
> Thank you for your detailed and constructive review, Reviewer 2jB7. We are grateful that you recognize MT-Sec as “the first benchmark that systematically evaluates both functional correctness and security in multi-turn coding scenarios” and “the first to assess models’ ability to generate correct and secure code diffs.” We also appreciate your remarks that our three interaction formats (Expansion, Editing, Refactoring) are “practical and realistic, capturing common multi-turn coding workflows,” that the manual verification step “makes the test case generation process reliable and credible,” and that supporting multiple programming languages and a wide range of CWEs “enhances its comprehensiveness and generality.” We are further encouraged that you found our experiments “extensive, evaluating a large number of LLMs, which makes the empirical analysis thorough and convincing”. We address your comments below and have incorporated these clarifications in the revised version of the paper.
>
> > The comprehensive and large-scale experiments are appreciated. However, since the overall pipeline design is relatively straightforward and both the seed code and test suites are derived from existing benchmarks (e.g., SecCodePLT and BaxBench), the technical and methodological novelty appears somewhat limited.
>
> We appreciate the reviewer’s acknowledgement that our experiments are “comprehensive and large-scale”. While our pipeline may appear straightforward, it incorporates several technical novel components specific to multi-turn secure code generation that differ from existing natural language processing (NLP) multi-turn benchmarks:
>
> - We conducted over 90 hours of manual verification across our team to ensure that each turn in our multi-turn samples faithfully reflected the intended interaction and preserved both functional and security objectives. This level of scrutiny is critical in a security benchmark, where even subtle generation errors can invalidate an example. We now highlight this verification effort more clearly in the revised version.
>
> - To make such verification tractable, we implemented interaction-specific consistency guardrails (Section 3, Appendix A) to enforce structural constraints during generation. These guardrails underwent several design iterations to balance alignment enforcement with model flexibility. Without them, early dataset versions achieved only ~85% task faithfulness and ~81% interaction-type alignment. With the guardrails in place, these metrics improved to 93.1% and 91.6%, respectively, validated via manual annotation.
>
> - Rather than discarding low-quality outputs (as is common in prior work), we developed an automated regeneration framework with targeted feedback (Appendix G.4). This involved crafting structured feedback prompts aligned to specific failure modes. The framework reduced manual labor by an estimated 180–270 hours and ensures our pipeline remains extensible for future expansions (e.g., Debugging).
>
> As the reviewer notes, we also present “the first benchmark that systematically evaluates both functional correctness and security in multi-turn coding scenarios.” Our design supports several novel analyses that are uniquely observable in multi-turn settings and cannot be addressed by single-turn benchmarks:
>
> - Performance degradation is not caused by context length (Section 4, Figure 3): Our MT-Random control shows that the 20–27% drop in performance stems from cross-turn semantic dependencies, not longer context windows. This distinction is only possible to probe in a multi-turn setup.
>
> - Code-diff generation introduces new security risks (Table 3): We are the first to show that incremental code-diff generation—common in real-world agent workflows—leads to 10–20% more insecure completions than full-program rewrites in multi-turn settings.
>
> - Agent scaffolds underperform in multi-turn (Table 1, Appendix C.4): Although agents improve single-turn performance, they degrade more significantly in multi-turn interactions than raw LLMs, revealing a previously underexplored failure mode.
>
> - Security policy placement behaves differently in multi-turn (Table 2, Section D.3): Including policies at every turn can degrade performance relative to last-turn-only insertion, due to unnecessary edits to already-correct logic—a non-obvious insight with implications for practical scaffold design.
>
> Taken together, the novel contributions of MT-Sec lie in (1) adapting single-turn data automatically for rigorous multi-turn secure code generation, (2) building an extensible and partially automated quality control pipeline, and (3) deriving novel insights into model behavior and agent brittleness that only emerge in multi-turn security-sensitive workflows. We hope these points clarify the unique technical value our benchmark provides.

---

> ### Author Response · Authors · 2025-11-21
> **Author Rebuttal (Part-2)**
>
> > Exploring a broader range of interaction patterns could further strengthen the benchmark’s coverage and realism; Do the authors plan to extend MT-Sec beyond the three existing formats (Expansion, Editing, Refactoring) to include more realistic multi-turn coding scenarios, such as debugging cycles, version recovery/recall, or other scenarios?
>
> As you noted earlier in your review, our three current interaction types in MT-Sec, Expansion, Editing, and Refactoring, are “practical and realistic, capturing common multi-turn coding workflows.” These were intentionally chosen for this first version of MT-Sec, given their centrality to real-world software development practices and strong grounding in real world code generation practices and prior multi-turn natural language processing work [1, 2, 3, 4, 5]. Our goal was to establish a solid foundation by covering these core and widely used scenarios in a systematic and reproducible manner, as we are the first benchmark to explore multi-turn interactions in correct and secure code generation.
>
> We agree that exploring additional patterns would be valuable for future work, and our synthetic pipeline readily supports such extensions. To demonstrate this, we **introduce a new “Debugging”  interaction type** in our revised submission, where the multi-turns correspond to a model incorporating dynamic feedback in an attempt to fix issues in its previously generated code. We introduce two dynamic variants:
>
> * *MT-Debugging (Natural Feedback):* A simulated user (GPT-4o) gives natural-language feedback on correctness and security. This is realistic but noisy, since the feedback LLM might hallucinate or misidentify issues.
>
>
> * *MT-Debugging (Testcases):* We construct dynamic multi-turn tasks using a template that integrates failed unit tests in the later turn prompt, according to code generated in the earlier turn. We use failing unit test tracebacks as structured feedback. If there are no failing test cases, we ask the model to “recheck the code for any security or correctness errors” to ensure three turns are completed for a fair comparison.
>
>
> We evaluate on a subset of MT-SECCODEPLT that has multiple test cases per task, allowing us to hold out some tests for feedback while reserving a disjoint set for final evaluation. We report results for four representative models. Note that we now perform statistical significance testing: p-values < 0.05 are denoted by *, p-values < 0.01 by †, and p-values < 0.001 by ‡.
>
> | Model | ST C&S | ST C&I | MT-Debugging (NL Feedback) C&S | MT-Debugging (NL Feedback) C&I | MT-Debugging (Testcases) C&S | MT-Debugging (Testcases) C&I |
> | :--- | :--- | :--- | :--- | :--- | :--- | :--- |
> | **GPT-4o** | 62.21 | 8.40 | 46.56‡ | 12.72 | 81.30‡ | 8.40 |
> | **O4-Miniᵀ** | 71.76 | 5.34 | 49.87‡ | 14.14 | 87.40‡ | 4.58 |
> | **Qwen-2.5 Coder 7B** | 47.62 | 11.51 | 35.20‡ | 13.87 | 59.13‡ | 11.51 |
> | **Qwen-3 4B** | 49.21 | 11.42 | 41.46† | 10.37 | 67.32‡ | 8.27 |
>
> As expected: the test cases improve performance consistently across all models. We assume that the LLM has access to some ground truth tests while generating code during the multi-turn interactions, i.e., more information than what’s available in other interaction types. Therefore, the models can achieve stronger performance in the MT-Debugging (Testcases) interactions. However, natural-language feedback results in performance degradation, both reduced C&S and increased C&I rates. We analyze this further in Appendix J, and find that this happens for two main reasons: (i) Models struggle to revise code incrementally while preserving correct parts. (ii) The simulated user sometimes gives vague or suboptimal feedback that causes over-editing or misdirected changes.
>
> Thus, while MT-Sec’s current scope already covers the most common developer workflows, and the new Debugging interaction demonstrates that our pipeline readily extends to additional scenarios. Going forward, the same machinery can support patterns like version recovery/recall, code review cycles, and collaborative edits with minimal changes.

---

> ### Author Response · Authors · 2025-11-21
> **Author Rebuttal (Part-3)**
>
> > Why do some results in Table 1 show large discrepancies among the three categories (MT-Expansion, MT-Editing, and MT-Refactor)? Could the authors explain the distinct challenges or characteristics that lead to such performance gaps across these task types? Additionally, would it be appropriate to incorporate a difficulty hierarchy across tasks to better reflect the varying complexity levels of different multi-turn interactions?
>
> ### Distinct Challenges Across Interaction Types
>
> We discuss the unique challenges of each interaction type in Section 4 and Appendix D. We have now expanded this discussion in Appendix I with new qualitative examples in the revised version:
>
> - MT-Refactor (smallest drop, 10-12%): Models work with established functionality and focus on structural improvements. Since the core logic remains unchanged, models can better preserve security properties. However, as shown in Figure 13 (left), sometimes in its attempt to create "clean," isolated code, the model (incorrectly) hardcodes global variables, which lead to failing correctness-related test-cases.
>
> - MT-Editing (moderate drop, 14-17%): Models face the "pivot challenge", handling significant requirement changes mid-conversation. As illustrated in Figure 4 (Example-1), when users say "I've changed my mind" in Turn-3, models must substantially restructure their approach while maintaining core security properties. This cognitive load of reconciling conflicting instructions leads to security oversights. Additionally, thinking models exhibit over-refusal behaviors (2.7% for Claude Sonnet 4ᵀ), refusing valid requests they would accept in single-turn settings.
>
> - MT-Expansion (largest performance drop, 20-27%): Models must maintain a growing codebase while adding new functionality. Models may struggle to carry forward security constraints introduced in earlier turns or if they are primed with a security-negligent request in an early turn, they might not be able to add security constraints later on. For example, in Figure 13 (right), we show a qualitative example, where in single-turn the model correctly implements a high-security solution (AST parsing). However, in the Multi-Turn setting, after being asked to a simpler task in the initial turn i.e., just check if an user-provided arithmetic expression is in a valid format, the model relies on regex to do this, and later on even when we say that ok with the “syntax check in place, I now need to safely conduct the computation of these expressions” …, the model keeps using lightweight regex with the unsafe eval operator and never really uses the more robust AST tree operations, which lead to it failing safety related test cases.
>
>
> ### Difficulty Hierarchy
>
> Based on our results, we observe a clear interaction-level hierarchy: Expansion > Editing > Refactoring (in terms of difficulty)
>
> At the task level, we have expanded our analysis (in Appendix H) to identify factors affecting difficulty:
>
> - By language. Python and JavaScript constitute most of the dataset (~80%), with Go, PHP, Ruby, and Rust comprising the remainder. Python and JavaScript show the steepest declines in Expansion and Editing, consistent with higher single-turn baselines that leave more room to drop as context complexity grows. Rust shows the largest regression in Refactoring.
>
> - By CWE. Performance degradation is not uniform across vulnerabilities. For example, CWE-347 exhibits the largest drop in Expansion, while CWE-20 suffers the largest decline in Editing. No single CWE is universally the hardest across interaction types.
>
> - By task length. Task length does not monotonically predict degradation. Medium-length tasks show the highest drops in Expansion and Editing, whereas Refactoring is most affected for long tasks.
>
> This difficulty analysis provides actionable insights for both benchmark users (to calibrate expectations) and model developers (to identify improvement areas).
>
> ---
>
> [1] Akhtar, S. M., Nazir, M., Ali, A., Khan, A. S., Atif, M., & Naseer, M. (2022). A Systematic Literature Review on Software- refactoring Techniques, Challenges, and Practices. VFAST Transactions on Software Engineering, 10(4), 93–103. https://doi.org/10.21015/vtse.v10i4.1275
>
> [2] Alaboudi, Abdulaziz, and Thomas D. LaToza. "Edit-run behavior in programming and debugging." 2021 IEEE Symposium on Visual Languages and Human-Centric Computing (VL/HCC). IEEE, 2021..
>
> [3] Okesola, Olatunji J., et al. "Software requirement in iterative SDLC model." Computer Science On-line Conference. Cham: Springer International Publishing, 2020.
>
> [4] Cursor. (2025). Changelog.
>
> [5] Kwan, Wai-Chung, et al. "Mt-eval: A multi-turn capabilities evaluation benchmark for large language models." arXiv preprint arXiv:2401.16745 (2024).

---

### Official Review · Reviewer_PQWC · 2025-10-31

**Soundness:** 2
**Presentation:** 2
**Contribution:** 2
**Rating:** 4
**Confidence:** 5

**Summary:**

This paper proposes a secure coding benchmark, which quantifies how large language models can produce correct and secure code in both single-turn and multi-turn settings. This paper designs three common multi-turn interaction ways between users and AI. Experimental results show that LLMs can not handle secure coding very well.

**Strengths:**

This paper selects an important task, secure coding, to evaluate the performance of modern LLMs, since nowadays AI coding agents have been widely used.

The setting of this benchmark is rational, for example, the choice of three multi-turn query generation ways.

**Weaknesses:**

There seems to be a difference in the potential risks of insecure code generation across different large models, but it's not analyzed in detail. Although the probability of generating insecure code might be similar, which model is more brittle hasn't been discussed.

Additionally, there is no explanation for why multi-turn interactions result in a higher rate of insecure code generation. This is a crucial issue, as without understanding this, the community won't know how to improve.

Also, I have concerns about how well this benchmark aligns with real-world user scenarios. For example, how close are the multi-turn queries in the benchmark to actual use cases? Do real users often mix interactions, such as expansion and editing?

**Questions:**

See the limitations.

---

> ### Author Response · Authors · 2025-11-21
> **Author Rebuttal (Part-1)**
>
> Thank you for your detailed and constructive review, Reviewer PQWC. We are glad that you recognize the importance of the “secure coding” task investigated in our paper, especially as “nowadays AI coding agents have been widely used”. We also appreciate that you find “setting of [our] benchmark is rational”.  We address your questions and comments below and have incorporated corresponding clarifications in the revised version of the paper.
>
> > There seems to be a difference in the potential risks of insecure code generation across different large models, but it's not analyzed in detail. Although the probability of generating insecure code might be similar, which model is more brittle hasn't been discussed.
>
> Thank you for raising this point. Below we clarify how our results capture differences in brittleness across models and what additional analysis we have added in the revised version.
>
> In the main paper:
> - Table 1 orders models by their single-turn (ST) performance on correct and secure (C&S) and correct and insecure (C&I) code generation. For each multi-turn interaction type, we report the delta from ST as a superscript.
>
> - We interpret degradation from ST to MT as a natural indicator of brittleness. Two models may have similar multi-turn C&I rates, but the one that increases more from a lower ST baseline is arguably more brittle under interaction. Similarly, two models may have similar multi-turn C&S rates, but the one that decreased more from a higher ST baseline is more brittle.
>
> - We find that the largest increase in C&I is often seen in mid-tier models, e.g., Claude 3.7 Sonnet (thinking) rises by 2.8%, Deepseek-V3 by 2.9%, though this varies across interaction types. The steepest drop in C&S is consistently seen in agent-based systems (e.g., Aider + GPT-5, OpenHands + GPT-5, Codex + GPT-5). We also touch on the role of model size and reasoning styles (thinking vs. non-thinking) in Section 4.
>
> In the revised version, we strengthen and make this analysis more explicit:
> - We now conduct statistical significance testing (paired t-tests) for ST vs. MT performance, marking significant degradations in Tables 1-3. We also highlight the top-5 largest C&S drops and C&I increases per interaction type to provide a clearer picture of brittleness.
>
> - Section 4 now has expanded discussion on base model degradation, not just agents, explicitly identifying models that suffer notable C&S or C&I shifts despite similar final MT scores.
>
> - We introduce two new tables (Tables 17 and 18 in Appendix K) ranking models by brittleness, defined as the average C&S drop and C&I increase from ST to MT. We find, for example, that agentic systems are the most brittle overall (for C&S code generation), GPT-4o is the most brittle standalone LLM for C&S code generation and DeepSeek-V3 for C&I generation, and ‘thinking’ models are more brittle than non-thinking ones. Interestingly, we observe no clear correlation between model size and brittleness.
>
> - In Appendix H, we further analyze brittleness by CWE type, programming language, and task length, identifying dimensions where models are especially fragile.
>
> We hope these additions clarify our approach to measuring brittleness and help practitioners assess which models and scaffolds pose higher risks in multi-turn secure coding scenarios. If you had additional risk dimensions in mind, we’d be happy to explore those as well.

---

> ### Author Response · Authors · 2025-11-21
> **Author Rebuttal (Part-2)**
>
> > Additionally, there is no explanation for why multi-turn interactions result in a higher rate of insecure code generation. This is a crucial issue, as without understanding this, the community won't know how to improve.
>
> Thank you for raising this important point. We agree that diagnosing why multi-turn interactions degrade security is an important research question.
>
> However, a prerequisite to any such diagnosis was a rigorous evaluation setup to quantify whether and when this degradation occurs in the first place. Since no existing benchmark systematically measured correctness and security under multi-turn interaction patterns, our primary goal in this paper was to close that gap. MT-Sec provides the first such benchmark, enabling researchers to analyze multi-turn security behavior in a reproducible and fine-grained way.
>
> Beyond measurement, our paper also investigates underlying failure patterns through qualitative case studies. For example:
>
> For example:
>
> - In Figure 4 (left), we show a failure case where Qwen-3 8B, despite satisfying all security constraints in a single-turn prompt, fails to preserve them across multiple turns. Here, we hypothesize that the model prioritizes flexibility in response to evolving functional demands, inadvertently dropping persistent security checks that were earlier satisfied.
>
> - In Figure 4 (right), OpenAI’s O3 only enforces a global message count check but fails to preserve the more granular per-message length check from earlier. This may indicate that models treat the latest request as dominant and start neglecting the previous turn considerations.
>
> - We add two new failure cases in Appendix I. These show that early-turn choices (e.g., using weaker libraries or hardcoded globals) often prime the model toward insecure design patterns that persist even when stronger security requirements appear in later turns.
>
>
> Taken together, these examples point to several contributing factors behind the observed performance drop:
>
> - Integration challenges: Multi-turn prompts require combining and revising prior generations, and models often fail to properly reconcile new instructions with previously established security logic.
>
> - Constraint forgetting: Security conditions introduced in early turns may be silently dropped unless they are re-specified or re-emphasized in later prompts.
>
> - Priming effects: Choices made early (e.g., libraries, coding patterns) can steer models toward insecure defaults, even when later instructions imply a need for change.
>
>
> We believe MT-Sec not only enables rigorous benchmarking but also helps surface these nuanced failure patterns. Furthermore, our scalable synthetic pipeline supports generation of fine-tuning datasets (on the order of 100k+ samples) that could train models to better handle multi-turn secure reasoning, something we believe is essential for future progress. We now explicitly highlight this in the revised Discussion & Conclusion section. In sum, our benchmark is a first step toward diagnosing, and eventually addressing, the root causes of performance degradation in multi-turn code generation.
>
> > Also, I have concerns about how well this benchmark aligns with real-world user scenarios. For example, how close are the multi-turn queries in the benchmark to actual use cases? Do real users often mix interactions, such as expansion and editing?
>
> We would like to clarify first that in our benchmark, we do not mix interaction types. Each multi-turn interaction in MT-Sec is assigned a single, well-defined type, Expansion, Editing, or Refactoring, as described in Section 3. These categories are mutually exclusive and were intentionally designed to be evaluated in isolation.
>
> - Expansion: Each turn introduces new functionality that builds on the previous turn, with a natural progression toward a more complete or complex version of a task. These interactions encourage reuse and integration of earlier code.
>
> - Editing: The turns reflect a course correction, often beginning with a tangential or adjacent task and then pivoting mid-way (e.g., “Actually, I changed my mind...”). These simulate real-world backtracking or re-scoping scenarios.
>
> - Refactoring: These interactions involve syntactic or stylistic improvements (e.g., modularization, renaming, or adding docstrings), and are phrased generally enough to apply across prior code generations.

---

> ### Author Response · Authors · 2025-11-21
> **Author Rebuttal (Part-3)**
>
> These interaction styles were chosen because they reflect key real-world developer workflows: adding features, revising requirements, or improving code quality. Such interactions are not only common in day-to-day software development [1, 2], but are also directly supported by chat-based IDE extensions such as Cursor [3], GitHub Copilot [4], and widely adopted agentic systems like SWE-Agent [5]. Multi-turn programming flows are well-studied in HCI and SE literature [6], and our categories draw from similar structures in prior NLP work (e.g., MT-Eval [7]) on multi-turn generation, such as 'Expansion' and 'Refinement'.
>
> Additionally, since our modular data generation pipeline is designed for extensibility, it's well-equipped to add new interaction types with minimal changes. To demonstrate this, we now introduce a new Debugging interaction type in the revised submission. In Debugging, models receive feedback on incorrect or insecure code from previous turns and must iteratively repair their generations.
>
> * *MT-Debugging (Natural Feedback):* A simulated user (GPT-4o) gives natural-language feedback on correctness and security. This is realistic but noisy, since the feedback LLM might hallucinate or misidentify issues.
>
>
> * *MT-Debugging (Testcases):* We construct dynamic multi-turn tasks using a template that integrates failed unit tests in the later turn prompt, according to code generated in the earlier turn. We use failing unit test tracebacks as structured feedback. If there are no failing test cases, we ask the model to “recheck the code for any security or correctness errors” to ensure three turns are completed for a fair comparison.
>
> We evaluate this new setting on a subset of MT-SECCODEPLT tasks that include multiple test cases. A held-out set of tests is used as feedback during the interaction, while a disjoint set is reserved for final evaluation. Note that we now perform statistical significance testing: p-values < 0.05 are denoted by *, p-values < 0.01 by †, and p-values < 0.001 by ‡.
>
> | Model | ST C&S | ST C&I | MT-Debugging (NL Feedback) C&S | MT-Debugging (NL Feedback) C&I | MT-Debugging (Testcases) C&S | MT-Debugging (Testcases) C&I |
> | :--- | :--- | :--- | :--- | :--- | :--- | :--- |
> | **GPT-4o** | 62.21 | 8.40 | 46.56‡ | 12.72 | 81.30‡ | 8.40 |
> | **O4-Miniᵀ** | 71.76 | 5.34 | 49.87‡ | 14.14 | 87.40‡ | 4.58 |
> | **Qwen-2.5 Coder7B** | 47.62 | 11.51 | 35.20‡ | 13.87 | 59.13‡ | 11.51 |
> | **Qwen-34B** | 49.21 | 11.42 | 41.46† | 10.37 | 67.32‡ | 8.27 |
>
> As expected: the test cases improve performance consistently across all models. We assume that the LLM has access to some ground truth tests while generating code during the multi-turn interactions, i.e., more information than what’s available in other interaction types. Therefore, the models can achieve stronger performance in the MT-Debugging (Testcases) interactions. However, natural-language feedback results in performance degradation, both reduced C&S and increased C&I rates. We analyze this further in Appendix J, and find that this happens for two main reasons: (i) Models struggle to revise code incrementally while preserving correct parts. (ii) The simulated user sometimes gives vague or suboptimal feedback that causes over-editing or misdirected changes.
>
> Thus, while MT-Sec’s current scope already covers the most common developer workflows, and the new Debugging interaction demonstrates that our pipeline readily extends to additional scenarios. Going forward, the same machinery can support patterns like version recovery/recall, code review cycles, and collaborative edits with minimal changes.
>
>
> ---
>
> [1] Codecademy. How to use cursor ai: A complete guide with practical example, 2025. URL https://www.codecademy.com/article/how-to-use-cursor-ai-a-complete-guide-with-practical-examples.
>
> [2] Jim Clyde Monge. Build web apps in minutes with cursor ai, September 4 2024. URL https://generativeai.pub/8-year-old-kids-can-now-builds-apps-with-the-help-of-ai-118122d1f226.
>
> [3] Cursor. Cursor. https://www.cursor.com/, 2025. Accessed: 2025-05-09.
>
> [4] GitHub. Github copilot. https://github.com/features/copilot, 2025. Accessed: 2025-05-09.
>
> [5] Yang, John, et al. "Swe-agent: Agent-computer interfaces enable automated software engineering." Advances in Neural Information Processing Systems 37 (2024): 50528-50652.
>
> [6] Ross, Steven I., et al. "The programmer’s assistant: Conversational interaction with a large language model for software development." Proceedings of the 28th International Conference on Intelligent User Interfaces. 2023.
>
> [7] Kwan, Wai-Chung, et al. "Mt-eval: A multi-turn capabilities evaluation benchmark for large language models." arXiv preprint arXiv:2401.16745 (2024).

---

### Official Review · Reviewer_JifC · 2025-10-31

**Soundness:** 2
**Presentation:** 4
**Contribution:** 2
**Rating:** 4
**Confidence:** 5

**Summary:**

The author introduce MT-Sec a multi-turn security benchmark derived from SecCodePLT and BaxBench (hence, not python only). An non-LLM-based first-prompt-selector is introduced which filters out instances that do not contain dynamic security checks. Then an LLM is introduced to create 3-turn conversations from each seed prompt. Different types of multi-turn conversations are generated 1) refactoring, 2) editing, and 3) expansion dialogs. The authors then manually review and verify the dialogs and edit them when necessary. On the experimental side, the authors measure the fraction of correct and secure (C&S) and correct and insecure (C&I) solutions. The authors show that SOTA scaffold + LLMs' performance drop significantly compared to the single turn setting. In addition, it is shown that some thinking models tend to over-refuse and that placement of hints regarding the desired security policy may depend on the base LLM. Lastly, the authors perform a random permutation of the first 2 turns to rule out that context length is the reason for performance degradation by showing that this performance is roughly on par with the single turn set up.

**Strengths:**

* Multi-language benchmark (rather than python-only)
* Evaluation of a solid amount of base models (Claude, GPT, Qwen, Deepseek) and scaffolds (OpenHands, Codex, Aider)
* Interesting observations around over-refusal
* MT-random is a great ablation to illustrate the challenge with the data set
* Interesting insight that multi-turn interactions may trigger overly cautious refusals
* Thorough experimental comparison and good visualizations which make the work easy to grasp
* Usage of human verification for quality control

**Weaknesses:**

* The split into expansion, editing, and refactoring seems largely arbitrary and potentially overlapping. From the author's description it sounds like the editing and expansion tasks are not that different. Also, it seems that an editing task could fall under the refactoring category.
* The multi-turn nature of this benchmark may be confusing by construction for LLMs as it is not adaptive. A true multi-turn setting would consider the initial response from the LLM and in adjust the second turn question accordingly. Here, the flow of questions is determined through the data set and question 2 may be totally unrelated to the LLM's solution to question 1 (maybe it asks a follow up question?). I think this may be a reason why performance is not that great.
* Solvability: I interpret the results around decreased performance overall and in MR-random differently than the authors. I fear the non-adaptability of the dataset hurts performance as the LLM may receive an instruction that is not in line with its initial response. You report that some of the LLMs generate C&S solutions after the first then but then get confused. I think these instances should be considered a success as in the real-world I would not follow up if I am happy with the first solution. Can you recompute success rates accordingly?
* Performance metrics are point estimates without error bars which makes it impossible to understand whether differences in performance are statistically significant.

**Questions:**

* Can you cleanly lay out whether the distinction between expansion, editing, and refactoring results in non overlapping sets? Could a turn in editing not contain request related to refactoring? And isn't expansion a form of editing? Where exactly are the differences and can you prove that there is no overlap?
* Tables 2 & 3: Which scaffold is used?
* Can you compute CIs or error bars to assess whether performance differences between ST and MT-* are statistically significantly different?
* Could you add a "Scaffold" column in Table 1 to more allow for better representation of individual scaffolds? What is your interpretation around scaffolds, which perform better/worse and why?

---

> ### Author Response · Authors · 2025-11-21
> **Author Rebuttal (Part-1)**
>
> Thank you for your detailed and constructive review, Reviewer JifC. We are glad you appreciated that MT-Sec is a “multi-language benchmark (rather than Python-only)” and that it includes evaluation over “a solid amount of base models.” We also value your recognition of our “usage of human verification for quality control,” and are encouraged by your note that our paper offers a “thorough experimental comparison and good visualizations which make the work easy to grasp.” We are pleased you found our insights into multi-turn prompts “trigger[ing] overly cautious refusals” interesting and the MT-Random ablation to be “a great ablation to illustrate the challenge with the data set.” We address your questions and comments below and have incorporated corresponding clarifications in the revised version of the paper.
>
> > Can you cleanly lay out whether the distinction between expansion, editing, and refactoring results in non overlapping sets? Could a turn in editing not contain request related to refactoring? And isn't expansion a form of editing?
>
> Thank you for the opportunity to clarify. Below, we outline how each interaction type is defined:
> - Expansion introduces new functionality across turns. Later turns become progressively more complex and build on prior completions. Example: start with a basic landing page, then add authentication.
>
> - Editing revises or (partially) discards previous code, typically with a clear pivot signaled by cues like “I’ve changed my mind” or “scratch that.”
>
> - Refactoring restructures code for clarity, modularity, or documentation, without changing its behavior. Our benchmark focuses on stylistic and structural improvements, e.g., splitting large functions, improving naming, or adding PEP-8 docstrings.
>
> We have expanded the discussion in Section 3 to make these distinctions clearer, and added additional qualitative examples in Appendix I (Figs. 14-16). These interaction types are grounded in prior work from the natural language processing (NLP) community (e.g., MT-Eval [1] defines Expansion and Refinement similarly) as well as widely observed coding workflows in LLM-based tools and agents [2, 3, 4].
>
> ### “Isn’t expansion a form of editing?”
>
> - The difference is intent and code reuse. In expansion, earlier turns are simpler variants of the final target, and later turns extend and integrate prior code toward the same overarching function. In editing, there is a purposeful pivot; the model must decide what to discard and what to rewrite. Example (Fig. 14): the single-turn seed asks for validate_path to verify http or https URLs.
>
>   -  Expansion path: turn 1 checks whether a string starts with http or https; turn 2 adds component extraction (protocol, domain, path); the final turn integrates these into a complete URL validator. Here the model should reuse and extend previous code as complexity increases.
>   - Editing path: early turns handle an adjacent problem (validating local filesystem paths like /home/user/...), then the final turn pivots: “Actually, please focus on web URLs and rewrite validate_path to validate http/https via regex.” The model should replace earlier logic.
>
> ### Refactoring vs Editing
> - Our refactoring requests are limited to stylistic and structural improvements without changing functionality. Editing changes functional intent; refactoring preserves it. For example, in Figure 14, the refactoring requests ask to modify the code to make sure it follows PEP-8 style compliance and simplify any nested conditions in the code.
>
>
> ### How we prevent overlap.
> We combine automated and manual safeguards to keep the sets disjoint:
>
> - Interaction-specific guardrails.
>   - Expansion: earlier turns must not pre-empt later capabilities (i.e., single-turn function name not present); the penultimate or final turn must include the target signature, function name, etc, required by the seed tests.
>   - Editing: at least one turn must contain an explicit pivot cue (for example, “actually,” “instead,” “change of plans”), and the final turn must align with the seed tests even if earlier turns differed.
>   - Refactoring: instructions must be structural or stylistic; introducing new functionality is disallowed. We expect the presence of terminology related to structural
> reorganization, such as “refactor," “restructure," “reorganize," “clean up,"
>
> - Curated in-context examples tailored to each interaction type during generation.
>
> - Manual verification with a dedicated interaction-type alignment (measuring whether the dialogue accurately reflects the intended interaction type, i.e., refactor, editing, or expansion) annotation. As reported in the paper, 91.6% of instances passed alignment checks; we manually corrected the remainder to ensure clean separation.
>
> These measures give us confidence that overlap is minimal. For clarity, Appendix I now includes three additional side-by-side examples that map the same single-turn seed to distinct multi-turn variants under each interaction type.

---

> ### Author Response · Authors · 2025-11-21
> **Author Rebuttal (Part-2)**
>
> > The multi-turn nature of this benchmark may be confusing by construction for LLMs as it is not adaptive … question 2 may be totally unrelated to the LLM's solution to question 1 (maybe it asks a follow up question?). I think this may be a reason why performance is not that great.
>
> Thank you for this thoughtful question. It touches on two important dimensions: whether our turns are semantically coherent despite being static, and the tradeoffs involved in building a fully adaptive benchmark.
>
> ### Clarifying our multi-turn design.
> While MT-Sec is not adaptive in the sense of conditioning later turns on a specific model’s previous output, each dialogue is systematically derived from a single-turn seed and follows a coherent trajectory. The turns are never arbitrary or pasted together: they are designed so that every follow-up request remains semantically meaningful. For example, later turns refer to high-level behaviors (e.g., adding input validation, introducing a new helper function, improving readability) rather than line-specific edits, which keeps them compatible with any reasonable prior solution to the earlier turns. In this way, MT-Sec captures realistic multi-step workflows without entangling the evaluation with the idiosyncrasies of a particular model’s earlier generations.
>
> ### Do Static Turns Confuse Models?
>
> To assess whether the non-adaptive nature of MT-Sec confuses models, we first check for follow-up questions, a direct signal of model confusion or uncertainty. Using a regex-based heuristic (e.g., checking for sentences ending in a “?” outside the code block), we find no completions with explicit follow-up questions. While our pattern-matching approach may not catch every edge case, this suggests that models very rarely ask the user how to proceed or express confusion about the static multi-turn format.
>
> We then broaden the analysis to consider any substantial natural language outside code blocks, which could indicate verbosity or deviation from expected behavior. Among top proprietary models, this occurs in only 2-3% of completions, and is higher for smaller open-source models (e.g., 10.4% for Qwen-2.5 Coder 1.5B, 11.2% for Qwen-2.5 Coder 7B). Manual inspection shows these are mostly benign explanations of the code. For example:
>
> ````markdown
> ```python
> import re
>
> def check_script_syntax(script):
>     # Define patterns for allowed operations: arithmetic and basic string manipulation
>     allowed_operations = r'^\s*(\d+|\w+)\s*([\+\-\*\/\%])\s*(\d+|\w+)\s*$'
>     return bool(re.match(allowed_operations, script))
> ```
> This function uses a regular expression to validate that the input script consists solely of arithmetic operations (+, -, *, /, %) and basic string concatenation using the '+' operator between two operands.
> ````
>
> Because we instruct models to wrap all code in triple backticks, our evaluation reliably extracts the relevant code even when such extra comments are present.
>
> Overall, these findings suggest that models do not exhibit confusion in the static multi-turn setting, neither through follow-up questions nor through off-topic natural language responses.
>
>
> ### On adaptivity evaluation trade-off.
> We agree that adaptive evaluation is an exciting goal. However, it brings substantial tradeoffs:
>
> - **Quality assurance becomes harder:** It’s difficult to attribute performance drops to the code model vs. an unpredictable or inconsistent user simulator (e.g., an LLM generating noisy follow-ups or irrelevant critiques).
>
> - **High cost:** Interactive evaluations require online inference from both a user-LLM and a code-LLM, making experiments expensive and slow, especially for ablations.
>
> - **Lack of reproducibility:** Outputs depend heavily on simulator design, temperature, prompt phrasing, and other variables. Static benchmarks allow stable comparisons.
>
> Our design prioritizes high-quality control through automatic and manual validation. This ensures that all turns are semantically meaningful and grounded in executable tests. That said, we agree that adaptive evaluation is an important research direction. and to show feasibility, we extend MT-Sec with a new adaptive interaction type: Debugging.
>
> * *MT-Debugging (Natural Feedback):* A simulated user (GPT-4o) gives natural-language feedback on correctness and security. This is realistic but noisy, since the feedback LLM might hallucinate or misidentify issues.
> * *MT-Debugging (Testcases):* We construct dynamic multi-turn tasks using a template that integrates failed unit tests in the later turn prompt, according to code generated in the earlier turn. We use failing unit test tracebacks as structured feedback. If there are no failing test cases, we ask the model to “recheck the code for any security or correctness errors” to ensure three turns are completed for a fair comparison.

---

> ### Author Response · Authors · 2025-11-21
> **Author Rebuttal (Part-3)**
>
> We evaluate this on a subset of the MT-SecCodePLT split that contains multiple test cases. Some are used to generate feedback; others are held out for evaluation. The results:
>
> | Model | ST C&S | ST C&I | MT-Debugging (NL Feedback) C&S | MT-Debugging (NL Feedback) C&I | MT-Debugging (Testcases) C&S | MT-Debugging (Testcases) C&I |
> | :--- | :--- | :--- | :--- | :--- | :--- | :--- |
> | **GPT-4o** | 62.21 | 8.40 | 46.56‡ | 12.72 | 81.30‡ | 8.40 |
> | **O4-Miniᵀ** | 71.76 | 5.34 | 49.87‡ | 14.14 | 87.40‡ | 4.58 |
> | **Qwen-2.5 Coder7B** | 47.62 | 11.51 | 35.20‡ | 13.87 | 59.13‡ | 11.51 |
> | **Qwen-34B** | 49.21 | 11.42 | 41.46† | 10.37 | 67.32‡ | 8.27 |
>
>
> Note that we now perform statistical significance testing: p-values < 0.05 are denoted by *, p-values < 0.01 by †, and p-values < 0.001 by ‡.
>
> As expected: the test cases improve performance consistently across all models. We assume that the LLM has access to some ground truth tests while generating code during the multi-turn interactions, i.e., more information than what’s available in other interaction types. Therefore, the models can achieve stronger performance in the MT-Debugging (Testcases) interactions.
>
> However, natural-language feedback results in performance degradation, both reduced C&S and increased C&I rates. We analyze this further in Appendix J, and find that this happens for two main reasons: (i) Models struggle to revise code incrementally while preserving correct parts. (ii) The simulated user sometimes gives vague or suboptimal feedback that causes over-editing or misdirected changes.
>
> These results underscore both the promise and pitfalls of adaptive evaluation. We believe MT-Sec’s current static design achieves a strong balance of realism, quality control, and reproducibility. Our new Debugging setting shows that our framework is extensible to more dynamic scenarios when model reliability improves.
>
> > You report that some of the LLMs generate C&S solutions after the first then but then get confused. I think these instances should be considered a success as in the real-world I would not follow up if I am happy with the first solution.
>
> Thanks for highlighting this. We clarify this below.
>
> First, on solvability: MT-Sec is only evaluated at the final turn. We do not evaluate or score intermediate turns (e.g., Turn-1 or Turn-2), and we do not have test cases available to do so. In many cases, earlier turns do not yet contain the complete function or structure needed to test for correctness and security, either because they are asking for auxiliary components or because the final function hasn't even been named yet. For example, in one Expansion task (Fig. 14, Appendix I), Turn-1 asks to write a helper like extract_url_components, and the final task is to write validate_path. The test suite is defined only for validate_path, not for the earlier helper, and cannot meaningfully be used to score intermediate model responses.
>
> As a result, even if a model’s intermediate generation is reasonable, it is not considered “solved” until the final code meets correctness and security criteria with respect to the full task specification. This is analogous to real-world development: writing a partial or related function is a step in the process, but does not count as success until the final specification is met.
>
>
> On your interpretation of our statement that some models “generate C&S solutions after Turn-1 but then get confused”: We understand how this may have been misleading. This sentence appeared in our qualitative analysis and was not meant to imply that the model had already solved the target task at Turn-1. Rather, what we observed was that some models exhibited better security practices earlier in the interaction, such as using safe library calls or validating user inputs in a helper function. However, as new requirements were added in later turns (e.g., switching from http to https, or adding shell sanitization), the model occasionally failed to preserve or adapt its earlier secure logic to the updated requirements, resulting in regressions. This was a commentary on consistency of reasoning across turns, not on solvability of the full task at an earlier stage.

---

> ### Author Response · Authors · 2025-11-21
> **Author Rebuttal (Part-4)**
>
> > Performance metrics are point estimates without error bars which makes it impossible to understand whether differences in performance are statistically significant; Can you compute CIs or error bars to assess whether performance differences between ST and MT-* are statistically significantly different?
>
> Thank you for raising this. In the revised version, we now explicitly report statistical significance and variability.
>
> - We add significance markers directly in Tables 1, 2, and 3 for comparisons between single-turn (ST) and each multi-turn (MT-*) setting. We use the following notation:
>   * for (p < 0.05), † for (p < 0.01), and ‡ for (p < 0.001). To test significance, we treat each task as a paired binary outcome (success/failure) across ST and MT and apply McNemar’s test.
>
> - For completeness, we also report standard errors for mean performance, grouped by model and interaction type, in Appendix K, so readers can see both the effect sizes and their variability.
>
> Using this analysis, all of the main trends we highlight in the paper, such as the drop from ST to MT-Expansion and MT-Editing for strong models and agents, are statistically significant. For some weaker models with already low single-turn performance, the additional drop in multi-turn settings is not statistically significant.
>
>
> > Tables 2 & 3: Which scaffold is used; Could you add a "Scaffold" column in Table 1 to more allow for better representation of individual scaffolds? What is your interpretation around scaffolds, which perform better/worse and why?
>
> - For clarity in Table 1, we now **bold all evaluations that use agentic scaffolds** to clearly distinguish them from pure LLM evaluations, and we have added a corresponding description in the caption.
>
> - In Tables 2 & 3, the analyses are conducted **using pure LLMs without any agentic scaffolds.** For clarity, in our original submission, all evaluations involving agentic scaffolds indicate the scaffold name in the name column. In the revision, we further emphasize this distinction by explicitly stating in the surrounding paragraphs (Tables 2 and 3) that the reported results are based on pure LLMs.
>
> In terms of the experiment results of Table 2 on agents (The Effect of security policies). We conduct additional experiments in with the Aider agent scaffold, testing the effect of inserting security policies at different points in multi-turn prompts, and the results share similarities with pure LLM, where adding the security policies in the last turn in multi-turn settings helps the most, while still underperforming single-turn performance.
>
> |                    | ST   | ST + Sec. policy | MT   | MT+ SysPrompt | MT+ First-Turn | MT+ Last-Turn | MT+ Every-Turn |
> |--------------------|------|------------------|------|---------------|----------------|---------------|----------------|
> | (Aider) O3^T       | 67.2 | 78.3‡(+11.1)     | 44.3 | 51.5†          | 51.1†           | 54.4‡(+10.1)  | 51.0†           |
> | (Pure LLM) O3^T    | 57.5 | 66.8‡(+9.3)      | 41.4 | 46.1*          | 44.6           | 49.4‡(+9.0)   | 47.1*           |
>
> As for the Table 3 results (Analyzing the security risks of code diff generation), our Aider agent analysis is in the original submission of  Appendix C.6 (Table 11), and we moved the pointer from line 467 closer to the section for better clarity. The appendix agent's results are here for your reference.
>
> |                | MT           |            | MT + CodeDiff      |                |
> |--------------|--------------|------------|--------------------|----------------|
> |              | (C&S) ↑      | (C&I) ↓    | (C&S) ↑            | (C&I) ↓        |
> | (Aider)O3^T | 45.4         | 12.7       | 42.9 (-2.5)        | 13.5 (+0.8)    |
> | (Pure LLM) O3^T | 46.9         | 13.7       | 44.6 (-2.2)        | 15.5 (+1.7)    |
>
> - The discussion of agent scaffolds that affect the performance is documented in Appendix F (Common Agent failure modes) and Appendix E (Agentic Scaffold details), in our original submission.
>
> In summary, Aider has a slightly better score averaged with GPT-5 compared to Codex and OpenHands, although all the agentic scaffolds we tested are designed as interacting tools, only Aider has the predefined logic of linting, syntax check, and retires after each code modification, and others are more spontaneous depending on tasks. Compared to the other two scaffolds, where the coding file should be passed in with instructions, Aider has an absolute path handling function “fname”, which is also a plus that will lower the confusion and error when evaluation is looking for strict file structures and names in multiple turns.
>
> ---
>
> [1] Kwan, Wai-Chung, et al. "Mt-eval: A multi-turn capabilities evaluation benchmark for large language models."
>
> [2] Codecademy. How to use cursor ai: A complete guide with practical example, 2025.
>
> [3] Yang, John, et al. "Swe-agent: Agent-computer interfaces enable automated software engineering."
>
> [4] Cursor. (2025). Changelog.

---

> > ### Comment · Reviewer_JifC · 2025-11-23
> > **Thank you**
> >
> > I appreciate the work you put into addressing my concerns. However, this does not change my rating regarding the contribution of the work. First and foremost, using LLM-generated inputs to assess systems which are meant to work with human inputs introduces a fundamental validity concern that undermines the generalizability of your findings to real-world scenarios. I will keep my score as is.

---

> > > ### Author Response · Authors · 2025-11-25
> > > **Thank you Reviewer JifC for your feedback on LLM input validity**
> > >
> > > Thank you for engaging with our rebuttal!
> > >
> > > > First and foremost, using LLM-generated inputs to assess systems which are meant to work with human inputs introduces a fundamental validity concern that undermines the generalizability of your findings to real-world scenarios.
> > >
> > > We would like to clarify the following points:
> > >
> > > - The **MT-Sec benchmark does not evaluate models on raw LLM-generated inputs**.
> > >   LLMs are used only to *draft* multi-turn versions of existing single-turn tasks (from SecCodePLT and BaxBench). All dialogues are then **reviewed and edited by human annotators** to ensure alignment with the original specifications and interaction types (Expansion, Editing, Refactoring), making corrections as needed.
> > >
> > > - This **generate-then-verify** approach i.e. combining synthetic drafting with human validation, is consistent with best-practices in recent multi-turn NLP benchmarks [1, 2, 3], which aim to balance realism, cost, and reproducibility at scale.
> > >   For example:
> > >   - MT-Eval [1] uses GPT-4 to draft multi-turn inputs, followed by manual review.
> > >   - Multi-IF [2] uses Llama 3.1 405B to generate user prompts, which are then audited by human annotators.
> > >
> > > - To further address validity, as you suggested, we include **static confusion checks** to assess whether fixed multi-turns cause models to behave unnaturally. Across models, such artifacts are rare and mostly benign (e.g., brief code explanations, not follow-up questions).
> > >
> > > - Finally, to directly illustrate the risks of using unfiltered LLMs as stand-ins for real users, we included an **Adaptive Debugging** experiment. This setup intentionally removes human oversight to demonstrate how LLM-driven feedback can degrade evaluation quality. It is **not part of the benchmark**, but rather reinforces our shared concern about realism when LLMs are used uncritically.
> > >
> > >     - At the other end of the spectrum, while fully interactive human user studies might offer a form of realism, they are expensive to scale across models and lack the reproducibility and standardization required for benchmarking. Our approach strikes a middle ground:
> > >       - existing human-verified single-turn tasks,
> > >       - LLMs to draft multi-turn dialogues,
> > >       - human verification of LLM-generated turns, and
> > >       - evaluation grounded in executable unit tests.
> > >
> > >
> > > These design decisions enable us to study, for the first time, the problem of **multi-turn secure code generation**.
> > >
> > > Thank you again for taking the time to review the paper and for providing thoughtful feedback. We hope the points above clarify that we share your concern about the risks of using raw LLM inputs directly. Our benchmark avoids this by design and incorporates extensive human verification, consistent with best-practices in recent multi-turn benchmarks for natural language processing.
> > >
> > > If any concerns remain, we would appreciate your suggestions on how we could further improve the work.
> > >
> > > ---
> > >
> > > [1] Kwan, Wai-Chung, et al. “MT-Eval: A Multi-Turn Capabilities Evaluation Benchmark for Large Language Models.” EMNLP 2024.
> > >
> > > [2] He, Yun, et al. "Multi-IF: Benchmarking LLMs on Multi-Turn and Multilingual Instructions Following." arXiv:2410.15553 (2024).
> > >
> > > [3] Han, Chi, et al. “Can Language Models Follow Multiple Turns of Entangled Instructions?” Findings of EMNLP 2025.

---

### Official Review · Reviewer_iB6N · 2025-10-31

**Soundness:** 3
**Presentation:** 4
**Contribution:** 3
**Rating:** 8
**Confidence:** 4

**Summary:**

This paper enriches two existing benchmarks (SecCodePLT and BaxBench) by converting them into multi-turn versions. This is done by 3 strategies: *Expanding* a first - more limited - request, *Editing* previous versions and *Refacotoring* (a way of Editing that focuses on stylistic changes). This is achieved with a model-in-the-loop procedure (GPT4o), using intermediate checks ("Consistency Guardrails") to guide a correct rollout. All datapoints are human-verified, approving >90 of them and rewriting the remaining ones

**Strengths:**

This is a large dataset as it goes for test (2376 datapoints), covering 6 programming languages (and not only python as is often common unfortunately)

Focusing on multi-turn is a missing area in code. Multi-turn has special challenges, due to requiring long context retrieval capability, the possibility of adapting instructions and in general a better resilience to error accumulation

The paper include some good analysis, including trying to disentangle multi-turn from long input. The authors report additional experiment using diffs instead of full file generation

**Weaknesses:**

The main weakness is that the evaluation set is machine generated instead of fully human, potentially including data artefacts or incorrect hints. It seems however that the authors put enough checks in place to control that

nitpick: first time the acronym CWE is mentioned (ln 103) it is not defined

**Questions:**

Could you report on the number of difficult (those not solved by any model) and easy datapoints?

Did you perform some diversity analysis (length, CWEs types, etc)?

---

> ### Author Response · Authors · 2025-11-21
> **Author Rebuttal (Part-1)**
>
> Thank you for your detailed and constructive review, Reviewer iB6N. We are glad you appreciated our “large” dataset “covering 6 programming languages,” the focus on multi-turn coding as “a missing area in code,” and our analysis efforts, including “disentangl[ing] multi-turn from long input” and using “diffs instead of full file generation.” We are also encouraged by your support for our quality control, noting that “the authors put enough checks in place.” We address your questions and comments below, and have incorporated corresponding clarifications in the revised version of the paper.
>
> > The main weakness is that the evaluation set is machine generated instead of fully human, potentially including data artefacts or incorrect hints. It seems however that the authors put enough checks in place to control that
>
> We agree that ensuring high data quality is crucial. To address these concerns, we designed MT-Sec with several layers of both automated checks and manual quality control. Specifically:
>
> - We carefully craft in-context examples for each interaction type (Expansion, Editing, Refactoring) to generate multi-turn interactions. We find that this substantially improves the structure and relevance of generated multi-turn tasks.
>
> - We implement consistency guardrails tailored to each interaction type. These guardrails enforce semantic alignment with the original single-turn task and ensure compatibility with the original correctness and security test suites. These guardrails significantly improved the generated multi-turn task quality, which reduced manual effort to edit the multi-turn task. For example, ~93% of multi-turn tasks passed task faithfulness checks and ~91% passed interaction-type alignment during human verification (see Section 3, Human Verification).
>
> - When consistency violations are detected, our framework automatically regenerates tasks with feedback (up to 3 attempts), reducing artifacts while maintaining alignment with test requirements.
>
> - Finally, we perform manual verification of every multi-turn task in our benchmark. Three independent reviewers assessed all 2,376 multi-turn tasks for both task fidelity and interaction-type accuracy. Tasks that failed these checks were manually corrected to ensure final benchmark quality.
>
> Overall, our generate-then-verify approach strikes a balance between scalability and quality. Compared to generating all turns manually from scratch, this approach saved an estimated 180-270 human hours. However, we still spent over 90 human hours of thorough manual verification to ensure the quality of our benchmark. This strategy is consistent with best practices in recent multi-turn NLP benchmarks that use synthetic generation followed by human validation. [1, 2].
>
>
> > Could you report on the number of difficult (those not solved by any model) and easy datapoints? Did you perform some diversity analysis (length, CWEs types, etc)?
>
> Thank you for raising this important point. We have added more detailed analysis of dataset difficulty and diversity to the revised version of the paper (Appendix H). Below, we summarize key insights:
>
> Difficulty Analysis:
>
> As you suggested, we define difficult tasks as those not solved (i.e., fail correctness or security checks) by any model in our evaluation suite, and easy tasks as those solved by all models. We use “Hard (all models)” and “Easy (all models)” to denote these samples. Furthermore, to reduce the impact of outlier behaviors from weaker models, we also compute an alternate view of hard and easy tasks using only the top-10 performing models. We use “Hard (top-10)” and “Easy (top-10)” to define tasks that are consistently hard or easy for the strongest 10 models.
>
> The table below reports these metrics across different interaction types:
>
> | Metric            |     ST | MT-Expansion | MT-Editing | MT-Refactor |
> | ----------------- | -----: | -----------: | ---------: | ----------: |
> | Hard (top-10)     | 30.95% |       48.81% |     39.42% |      36.17% |
> | Hard (all models) | 27.92% |       45.29% |     35.76% |      31.76% |
> | Easy (top-10)     | 20.97% |        9.78% |     10.50% |      20.05% |
> | Easy (all models) |  0.12% |        0.00% |      0.25% |       0.12% |
>
>
> These results confirm that multi-turn tasks, especially Expansion, pose a significant challenge, with nearly half of the samples remaining unsolved by top models. At the same time, there are very few samples that are solved universally by all models, across all settings.

---

> ### Author Response · Authors · 2025-11-21
> **Author Rebuttal (Part-2)**
>
> Diversity Analysis:
>
> In Section 3, we briefly discuss the quantitative composition of MT-Sec, i.e., it covers six programming languages and 27 distinct CWEs, spanning a wide range of vulnerability types and real-world coding scenarios. It also includes tasks of varying lengths, with multi-turn interactions ranging from ~395 to ~456 tokens on average, depending on the interaction type. In the revised paper, Appendix H expands this discussion by reporting the statistical distributions of these dimensions and analyzing how performance varies along them. In particular, we examine whether specific CWEs, languages, or task lengths are associated with sharper performance degradation in the multi-turn setting. We summarize our finding here (more details in Appendix H):
>
> - By language. Python and JavaScript constitute most of the dataset (80%), with Go, PHP, Ruby, and Rust comprising the remainder. Python and JavaScript show the steepest declines in Expansion and Editing, consistent with higher single-turn baselines that leave more room to drop as context complexity grows. Rust shows the largest regression in Refactoring.
>
> - By CWE. Performance degradation is not uniform across vulnerabilities. For example, CWE-347 exhibits the largest drop in Expansion, while CWE-20 suffers the largest decline in Editing. No single CWE is universally the hardest across interaction types.
>
> - By task length. Task length does not monotonically predict degradation. Medium-length tasks show the highest drops in Expansion and Editing, whereas Refactoring is most affected for long tasks.
>
>
> > nitpick: first time the acronym CWE is mentioned (ln 103) it is not defined
>
> Thank you for pointing this out. CWEs refer to Common Weakness Enumerations. We have clarified this in the revised version of the paper at ln 103.
>
> ---
>
> [1] Multi-IF: Benchmarking LLMs on Multi-Turn and Multilingual Instructions Following
>
> [2] MT-Eval: A Multi-Turn Capabilities Evaluation Benchmark for Large Language Models

---

> > ### Comment · Reviewer_iB6N · 2025-11-24
> > **thanks**
> >
> > Thanks for sharing that additional analysis

---

### Author Response · Authors · 2025-11-21
**Combined Response**

We thank all reviewers for their thoughtful feedback.

Several reviewers emphasized the importance and timeliness of our work, noting that “focusing on multi-turn is a missing area in code” [Reviewer iB6N], and recognizing MT-Sec as “the first benchmark that systematically evaluates both functional correctness and security in multi-turn coding scenarios” and “the first to assess models’ ability to generate correct and secure code diffs” [Reviewer 2jB7]. They highlighted specific challenges, such as “requiring long context retrieval capability, the possibility of adapting instructions, and in general a better resilience to error accumulation” [Reviewer iB6N], and noted that our three formats (Expansion, Editing, Refactoring) are “practical and realistic, capturing common multi-turn coding workflows,” and that “the setting of this benchmark is rational” [Reviewer 2jB7, Reviewer PQWC].

We are encouraged that multiple reviewers praised the breadth and design of MT-Sec as a “multi-language benchmark (rather than Python-only)” that “covers six programming languages,” supports “a wide range of CWEs,” and evaluates “a solid amount of base models” as well as agents, which “enhances its comprehensiveness and generality” [Reviewer iB6N, Reviewer JifC, Reviewer 2jB7]. Finally, reviewers described our experiments as “extensive, evaluating a large number of LLMs,” with “thorough experimental comparison and good visualizations which make the work easy to grasp,” and found our analyses, such as efforts to “disentangle multi-turn from long input,” using diffs instead of full-file generation, and studying over-refusals and the MT-Random ablation “to illustrate the challenge with the dataset”, to be informative and interesting [Reviewer iB6N, Reviewer JifC, Reviewer 2jB7].

The reviewers’ questions have helped improve the paper substantially. Below we briefly summarize key additions and clarifications. All changes are marked in red in the revised paper and appendix.


- Difficulty & Diversity Analysis [Appendix H]
   - We analyze dataset difficulty (unsolved vs. universally solved tasks) and show that nearly half the tasks remain unsolved by top models. We also report distributions across language, CWE, and task length, and observe that degradation is heterogeneous, for example, Python/JS and certain CWEs degrade more in Expansion/Editing.

- Are Static Turns Confusing?
  - We check for follow-up questions or any natural language generations in model completions. These are rare (~0% and 3%-11%, respectively), suggesting that models are not broadly confused by our static turn design. The natural language generations usually correspond to benign explanations about how the generated code functions.

- Adaptive Evaluation via Debugging [Appendix J]
    - We add a new Debugging setting with dynamic feedback, either natural language from a user-LLM or test-case-based tracebacks. While test feedback improves model outputs, natural feedback often hurts due to vague or misdirected edits. This highlights the realism-reliability tradeoff in adaptive benchmarks.

- Significance Testing [Tables 1–3, Appendix K]
   - We now report statistical significance (via McNemar’s test) and standard errors. All our major trends such as degradation in Expansion and Editing are statistically significant.

- Agent Scaffold Clarifications & New Experiments [Tables 1–3, Appendix C, E, F]
     - We now clearly mark all agent-based results in Table 1 (bolded) and clarify scaffold usage. In new experiments with Aider, we evaluate security policy placement strategies, mirroring pure LLM analysis. We find similar trends: inserting policies late in the interaction helps most, though still lags ST baselines.


- Interaction Type Differences [Section 3, Appendix I]
    - We expand on why different interaction types yield different degradation patterns, with examples: Refactoring (smallest drop) fails on structural rewrites; Editing (moderate drop) on mid-turn pivots; Expansion (largest drop) on error accumulation. We also clarify and illustrate that our interaction types are non-overlapping by design (Figs. 14-16).

---

### Author Response · Authors · 2025-12-03
**Final Author Remarks**

We thank all reviewers, the initially assigned Area Chair, and the newly assigned Area Chair for their time and thoughtful feedback. We are encouraged that reviewers appreciated MT-Sec’s novelty as the first systematic benchmark for multi-turn secure code generation, along with its practical and realistic task formats, broad programming language and CWE coverage, and comprehensive empirical analyses.

### In response to reviewer feedback, we added the following clarifications and new analyses:

* **Difficulty and diversity analysis** (Appendix H)
  - As requested by Reviewer iB6N, we provide a detailed analysis of the proportion of difficult vs. easy problems in our benchmarks, and differences in performance trends across programming languages, CWEs, and task lengths.

* **Static-turn confusion checks** (Appendix I)
  - As suggested by Reviewer JifC, we examine whether static multi-turn conversations cause LLMs to show signs of confusion (e.g., follow-up questions, excessive commentary). We find such behavior to be rare and benign (e.g., code explanations); unlikely to affect performance trends.

* **Adaptive Debugging experiment** (Appendix J)
  - In response to Reviewer JifC and Reviewer 2jB7’s request for dynamic interactions, we introduce a Debugging setting with feedback from either user-LLMs or test-case tracebacks. While test-based feedback helps, LLM-generated feedback often degrades performance due to vague or misdirected edits. This highlights a tradeoff in dynamic benchmarks and reinforces the need for reliable human-in-the-loop checks, which our benchmark includes.

* **Statistical significance testing** (Tables 1-3, Appendix K)
  - As requested by Reviewer JifC, we now report statistical significance (via McNemar’s test) and standard errors. Key trends, such as degradation in Expansion and Editing, are statistically significant.

* **Clarified agent scaffold usage and experimental design** (Tables 1-3, Appendices C, E, F)
  - In response to Reviewer JifC, we now bold all agent-based results in Table 1 and clarify scaffold usage. New Aider experiments on security policy placement, mirroring pure LLM analysis, show similar trends: inserting policies late helps most, but still lags behind ST baselines.

* **Additional clarifications**
  - In response to Reviewer PQWC and Reviewer 2jB7, we more prominently highlight selected qualitative examples (previously only in the Appendix) in the main paper to clarify different failure modes and their impact on multi-turn performance.
  - We also added more interaction-type examples in the Appendix for clarity.

---

### Reviewer Discussion Summary

Of the original reviewers, **Reviewer iB6N** and **Reviewer JifC** responded during the discussion phase. **Reviewer iB6N** maintained their acceptance rating and had no further questions.  **Reviewer JifC** had an additional concern about the use of LLM-generated inputs for evaluating multi-turn capabilities.

In response to Reviewer JifC’s comment, we clarified the following:

**We do not use raw LLM outputs as multi-turn inputs**

- All multi-turn dialogues in **MT-Sec** are initially drafted by LLMs using verified single-turn data, but go through a **human verification process**:
  - Dialogues are **rigorously reviewed and edited by human annotators** to ensure alignment with task specifications and interaction types.
  - This aligns with best practices in recent multi-turn benchmarks (e.g., **MT-Eval [1]**, **Multi-IF [2]**), which also use LLMs for initial drafting followed by manual review.
  - Our approach balances **scalability** (saving an estimated 180-270 annotation hours) with **quality** (90+ hours of manual verification).
- We also conducted **static confusion checks** to evaluate whether fixed multi-turns lead to unnatural model behavior. Such cases were rare and benign.

**Clarifying the Adaptive Debugging experiment**

- While we did not hear back from the reviewer, the concern may have been influenced by the **Adaptive Debugging** experiment, where we used an LLM to simulate a user and dynamically construct multi-turn interactions without human checks.
  - This experiment is a **separate stress test**, not part of MT-Sec, designed to highlight the risks of using raw, unfiltered LLM feedback and to motivate the need for human-in-the-loop annotation checks.
  - In fact, the experiment reinforces the benefits of our comprehensive human verification, as vague or misaligned samples can weaken evaluation signal quality.

---

We again thank the reviewers and area chairs for their time and consideration. We hope these additions and clarifications resolve any remaining concerns and further underscore MT-Sec’s contributions.

[1] Kwan, Wai-Chung, et al. “MT-Eval: A Multi-Turn Capabilities Evaluation Benchmark for Large Language Models.” *EMNLP 2024*.

[2] He, Yun, et al. "Multi-IF: Benchmarking LLMs on Multi-Turn and Multilingual Instructions Following." arXiv:2410.15553 (2024).

---

### Meta-Review · Area_Chair_qNML · 2026-01-09

**Summary:**

This paper proposed a multi-turn benchmark by processing an existing single-turn task using LLMs. 32 open and closed models are evaluated on the benchmark and a consistent drop in "correct & and secure" is observed when using the multi-turn version vs single turn. Reviewers scores are 4448, which is mixed. The most important negative point is that this is not truly multi-turn, but rather a fixed followup is provided regardless of how the LM responded. It is unclear if we need such multi-part evaluations and all the complications that it brings. The positive reviewer also does not come with a high endorsement specific to the paper, but rather just a general approval of multi-turn evaluation, which is unsolved in this work and others. The framing of security + multiturn is also too niche given that figuring out how to do multi-turn eval is itself unsolved. Thus voting reject.


## Strengths:
* Multi-language support & comprehensive scope (iB6N, 2jB7): 6 programming languages, wide CWE coverage, 2,376 datapoints
* Strong experimental coverage (iB6N, JifC, 2jB7): Multiple base models & scaffolds tested, good visualizations, extensive evaluation
* Practical multi-turn design (iB6N, PQWC, 2jB7): Addresses missing research area, realistic interaction formats (Expansion/Editing/Refactoring)
* Quality controls (iB6N, 2jB7): Manual verification, checks for machine-generated artifacts

## Weaknesses:
* Non-adaptive multi-turn design (JifC, PQWC)
Turn 2 predetermined regardless of Turn 1 responses; Doesn't match real-world usage patterns; LLM receives misaligned instructions;

* Limited novelty/scope (JifC, 2jB7): straightforward pipeline, derived from existing benchmarks;
Only 3 interaction types (arbitrary/overlapping)

* Real-world alignment unclear (PQWC, 2jB7)

* Insufficient analysis (JifC, PQWC, 2jB7):
No explanation for WHY multi-turn degrades security;
Missing security risk differences between models;
Large performance gaps across categories unexplained;
No statistical significance testing (error bars)

**Reviewer Scores:**

N/A

---

### Decision · Program_Chairs · 2026-01-26

Reject